# Towards an AI-Based In-Bed Posture Detection System for Pressure Injury Prevention

Lindsay Stern
*Insitute of Biomedical Engineering; KITE Research Institute*
*University of Toronto; Toronto Rehabilitation Institute*
Toronto, Canada
lindsay.stern@mail.utoronto.ca

Atena Roshan Fekr
*Insitute of Biomedical Engineering; KITE Research Institute*
*University of Toronto; Toronto Rehabilitation Institute*
Toronto, Canada
atena.roshanfekr@uhn.ca

*Abstract*— **Pressure injuries (PIs) are common wounds among patients with decreased mobility who are unable to periodically redistribute their body weight. The most common technique to prevent PI development is through frequent repositioning, often requiring support from caregivers, which can be a costly and laborious task. Therefore, this paper investigates the use of a pressure sensitive sheet to automatically capture in-bed body postures to prevent PI development. Five Neural Networks were evaluated to classify 10 sub-postures using pressure distribution images. Two techniques were explored: directly classifying all 10 postures, and a hierarchical architecture. Although the hierarchical architecture with the ShuffleNet algorithm achieved the highest F1-Scores of 99.75% ± 1.43% for holdout (20% test set) and 93.53% ± 7.37% for Leave-One-Subject-Out (LOSO) cross-validation, direct classification provides more stable results. These results suggest that this approach has promising potential to detect common sub-postures and could be used to remind caregivers to facilitate timely repositioning, thereby preventing PI development.**

*Keywords—Posture Detection, Classification, Pressure Injury, Convolutional Neural Networks, Vision Transformer*

## I. INTRODUCTION

In-bed posture monitoring has become a prevalent area of research to help minimize the risk of Pressure Injury (PI) development. Pressure injuries can begin developing within minutes due to prolonged body-weight forces, typically occurring along the bony prominences of the body [1]. This continuous applied force can create cellular membrane breakage, initiating a cycle of cell death, inflammation, and ischemia, which can result in tissue injury [1]. Superficial wounds may also be present due to shear forces and moisture along the skin [1]. In the US, it is estimated that 3 million adults acquire PIs in the hospital, costing an estimated range between $500-$70,000 USD per patient [2]. Additionally, 60,000 US patients die each year due to PI complications [2]. In Canada, approximately 26% of patients across all healthcare settings suffer from PIs, costing between $1,247-$597,363 CAD per patient [3]. More specifically, 15% of elderly patients develop PIs within the first week of stay at the hospital and within the first four weeks of stay in long-term care facilities [4].

While repositioning patients every two hours is a crucial element of PI prevention, it can be challenging to integrate this practice seamlessly into busy healthcare environments [1] [5] [6]. Therefore, to ensure patients are receiving proper care and to alleviate some of the workload on the healthcare workers, an in-bed posture monitoring system needs to be developed. In this paper, we introduce a new system that uses a pressure mat, placed underneath bedsheets, and deep learning models to capture and classify real-time in-bed postures via pressure distribution images.

We use Convolutional Neural Networks (CNNs) and a Vision Transformer (ViT) to analyze pressure distribution images, aiming to classify these images into 10 distinct sub-postures. We explore two approaches: directly classifying all sub-postures (Approach 1) and a hierarchical architecture (Approach 2), which first classifies high-level positions (supine, right lateral, and left lateral) followed by a sub-posture classification within each high-level category. Both approaches are evaluated using holdout and Leave-One-Subject-Out (LOSO) cross-validation techniques.

The subsequent sections are organized as follows. In Section II, we describe previous works in this field. Section III discusses the dataset used and the proposed algorithms for the in-bed posture classification. In Section IV, we present the experimental results along with related observations and limitations. Lastly, Section V summarizes the concluding remarks.

## II. RELATED WORKS

Currently, devices, such as video infrared cameras or wearable technologies, have been developed to monitor in-bed postures and their corresponding durations [5]. However, there are some limitations to these devices. For example, video infrared cameras can be susceptible to environmental changes, such as the motion of a blanket, and have associated privacy concerns [5]. Similarly, wearable technologies, such as rings and wristbands, can obstruct sleep, reducing sleep quality, and are sensitive to motion artifacts [7]. The use of wearable e-textile sensors such as smart shirts [8] and underwear [9], for PI monitoring will introduce new challenges for care providers, as additional layers between the patient and the bed could increase the risk of PI development [10]. Therefore, there is a need for a privacy preserving and unobstructive system to monitor patients' in-bed body postures and prompt caregivers to reposition the patients if necessary. Pressure mats have gained traction as they offer an unobstructive and privacy preserving method to accurately detect in-bed postures that allow patients to move freely.

There are a variety of studies that have investigated in-bed posture detection using a smart mat composed of either pressure sensors or force sensors. Many of these studies only classified the main three or four in-bed postures, which include the supine, right lateral, left lateral, and sometimes prone postures [11] [12]. However, there are a few studies that classified multiple sub-postures correlated to the

positions listed above, which can provide the caregiver with a better understanding as to which areas along the body require offloading. Pouyan *et al.* conducted an experiment in 2013 on 20 subjects to classify eight different sleeping postures: supine, supine hands-on body, supine folded legs, supine crossed legs, right yearner, right fetus, left yearner, and left fetus [13]. This study used a commercially available pressure mat composed of 2,048 pressure sensors to capture these eight sub-postures. Using a K-Nearest Neighbour (KNN) classification algorithm, the study achieved an accuracy of 97.10% using a 10-fold cross validation technique [13]. Hu *et al.* conducted a study in 2021 on five subjects to classify six different sleeping postures: supine, log, right yearner, right fetus, left yearner, and left fetus [14]. This study used a pressure mat composed of 1,024 pressure sensors to capture in-bed sub-postures. A Convolutional Neural Network (CNN) algorithm was used to classify these six postures, resulting in an accuracy of 91.24% using a 20% test set holdout cross validation technique [14]. Xu *et al.* conducted an experiment in 2015 on 14 subjects using a pressure sensitive smart sheet composed of 8,192 pressure sensors [15]. Combining the body-earth mover's distance with a KNN algorithm to classify six postures, the study achieved an accuracy of 90.78% using LOSO cross validation [15]. Although these studies achieve high accuracies in classifying various sub-postures, none have evaluated the use of deep learning models to improve the classification performance, especially for LOSO cross validation. The LOSO cross validation technique is a necessary tool to evaluate the performance of classification models as the application of these models is for new and unseen users. Previous literature showed that most of the time the performance of algorithms drops when LOSO is used compared to the 10-fold or holdout cross validations [16] [17]. In this paper, we will evaluate the performance of various pre-trained deep learning models in classifying 10 different in-bed body postures using LOSO cross validation, and a holdout method with 20% of the data as the unseen test set.

## A. Dataset

The dataset used in this paper was collected by Pouyan *et al.* and made available through PhysioNet [18]. This dataset was collected in 2017. This dataset consists of two sections: the first section was collected on the Vista Medical FSA SoftFlex 2,048 pressure mat which was placed on top of a regular home mattress and the second portion of the data was collected on the Vista Medical BodiTrak BT3510 pressure mat that was placed on top of a sponge and air mattress [18]. In this paper, we have only used the first section of the dataset to classify in-bed sub-postures. The pressure data was collected with a sampling rate of 1.7 Hz. This data was reconfigured into pressure distribution images for each posture and subject. 13 subjects were evaluated, each lying in 17 different postures. Table I shows the subject-specific demographic information. Of the 17 postures, nine are in the supine position, four are in the right lateral posture, and four are in the left lateral position. Fourteen of these in-bed postures were recorded on a flatbed (zero incline), whereas

three of the supine postures were recorded at different bed inclines. To ensure consistency within our analysis, the three inclined supine postures were excluded. Due to the similarities between the Log 30° and Log 60° postures in the right and left lateral positions, which only differ slightly in torso angle, we merged these two classes into a single posture called "Log". The Bent knee and Fetus postures only had a small variation in torso and leg angles, merging these two postures into a single class called "Bent Knees". Fig. 1 displays sample frames correlated to the sub-postures within the supine, right lateral, and left lateral classes.

TABLE I. SUBJECTS' DEMOGRAPHIC INFORMATION

| Subject ID | Age | Height (cm) | Weight (kg) |
|---|---|---|---|
| 1 | 19 | 175 | 87 |
| 2 | 23 | 183 | 85 |
| 3 | 23 | 183 | 100 |
| 4 | 24 | 177 | 70 |
| 5 | 24 | 172 | 66 |
| 6 | 26 | 169 | 83 |
| 7 | 27 | 179 | 96 |
| 8 | 27 | 186 | 63 |
| 9 | 30 | 174 | 74 |
| 10 | 30 | 174 | 79 |
| 11 | 30 | 176 | 91 |
| 12 | 33 | 170 | 78 |
| 13 | 34 | 174 | 74 |

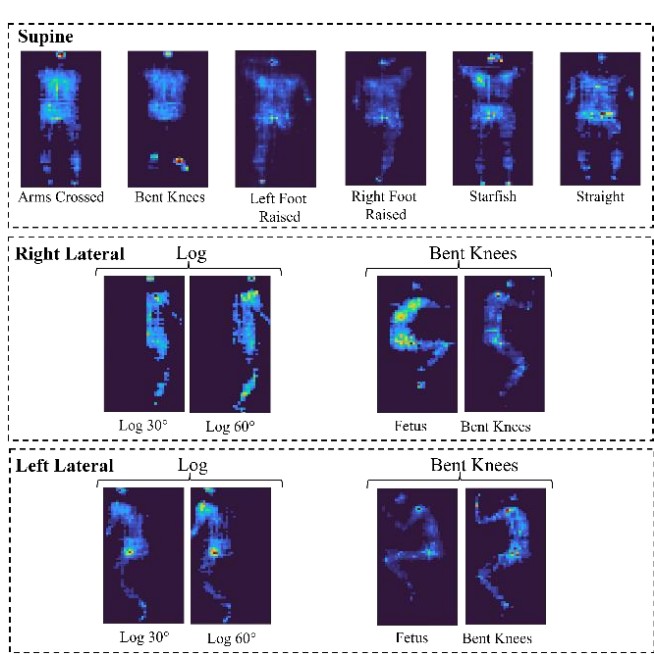

Fig. 1. Sample frames of the sub-postures pertaining to the supine, right lateral, and left lateral classes (14 classes on zero incline).

## B. Preprocessing

The dataset provided from PhysioNet was initially formatted in text files pertaining to each posture for each participant, where each row corresponded to a frame of data. This data was then preprocessed, where it was reconfigured into pressure distribution images with a size of $224 \times 224$ pixels.

## C. Deep Learning Algorithms

Four pre-trained 2D CNNs, AlexNet, GoogLeNet, ResNet-18, and ShuffleNet, and a Vision Transformer - Large (ViT-Large) were used to classify the in-bed postures. All four CNNs have been pre-trained on the ImageNet database composed of over a million images correlated to 1,000 object categories, such as a keyboard, a coffee mug, and various animals. AlexNet shown in Fig. 2 (a) is composed of eight layers in total, including five convolutional layers, two fully connected hidden layers, and one fully connected output layer [19]. Additionally, AlexNet uses the ReLU activation function, which involves a simple computation and facilitates easier model training [19]. GoogLeNet shown in Fig. 2 (b) is composed of 22 layers, including three convolutional layers, nine inception blocks, and one fully connected output layer [20]. Each inception block is composed of four parallel branches which outputs a concatenated value of the four branches [20]. ResNet-18, shown in Fig. 2 (c), is composed of 18 layers, including 17 convolutional layers, one fully connected layer, and one SoftMax layer [21]. Similar to AlexNet, ResNet-18 uses the ReLU activation function, with the option of a skip ReLU between each convolutional block if the block does not provide additional useful information to the model [21]. ShuffleNet, shown in Fig. 2 (d), is composed of 50 layers, using pointwise group convolution and channel shuffle to reduce computational costs while maintaining accuracy [22]. The ViT-Large, shown in Fig. 2 (e), was pretrained on the ImageNet-21k database and fine-tuned on the ImageNet database. This is a self-attention algorithm which divides an image into patches and embeds them linearly into a Transformer encoder, which includes multi-head attention and multi-layer perceptron (MLP) blocks, to effectively model long range interactions within images [23].

Transfer learning was used within all algorithms to apply the architect of those algorithms to the in-bed posture data. In total, 2,912 images were used to train and validate all models: 1,248 in the supine class, 832 in the right lateral class, and 832 in the left lateral class. These models were evaluated using both holdout and LOSO cross validation techniques. For our purposes, when using the holdout cross validation technique, we held out 20% of the data as a test set and trained the algorithms on the remaining 80% (train and validation sets). With this 80% of data, we used a 5-fold cross validation technique, where the algorithm separates the data into five folds, training on four of the folds and validating on the fifth fold. This was repeated five times to discover the best hyper-parameters. Once the algorithm is trained on 80% of data, we then tested it on the unseen 20% of data to understand its performance. The LOSO cross validation technique consists of training the algorithm on all participants' data except for one participant, and then testing the algorithm on that remaining unseen participant's data. This gets repeated several times until each participant is 'left out'. For example, in our dataset, the data from the first 12 of the 13 subjects was used to train the algorithm, with participant 13's data used as the test set. This would be repeated 13 times to complete the validation with all participants.

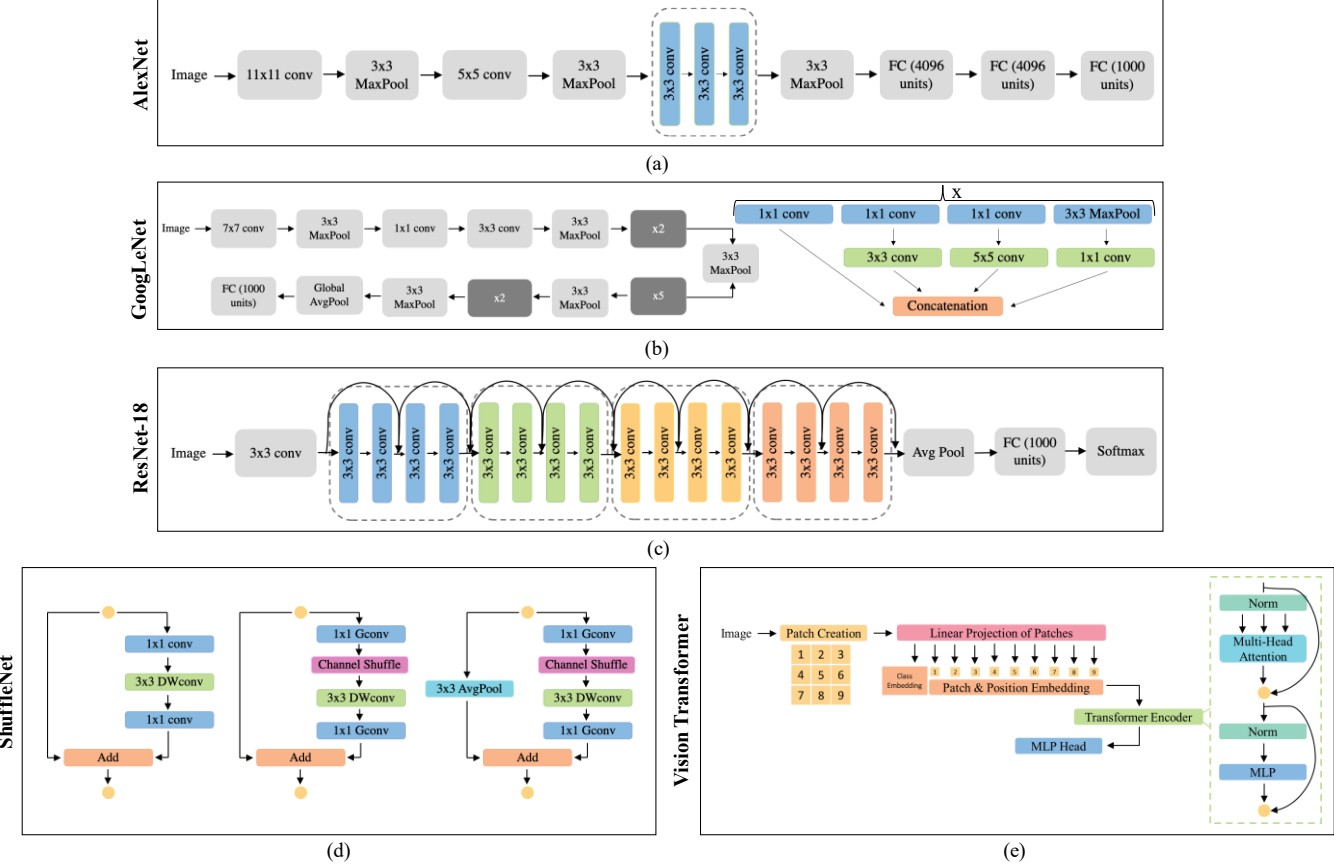

Fig. 2. Algorithmic architectures of (a) AlexNet [19], (b) GoogLeNet [20], (c) ResNet-18 [12], (d) ShuffleNet [22], and (e) Vision Transformer [23].

## D. Direct Classification and Hierarchical Architecture

Two different methods were used to classify the 10 postures: a direct classification (Approach 1) and a hierarchical architecture (Approach 2), shown in Fig. 3. Approach 1 is a single-step classification model, which directly classified all 10 sub-postures as 10 separate classes, creating an inter-class comparison. Approach 2 is a two-step classification model, using a multi-algorithmic approach, inspired by a previous study in [11] that obtained over 98% accuracy for LOSO cross validation. In stage I, a single algorithm classifies all images into one of 3 high-level categories (supine, right lateral, and left latera). In stage II, three algorithms—one for each high-level class—are used to further classify sub-postures within their respective categories, enabling detailed, within-class comparison.

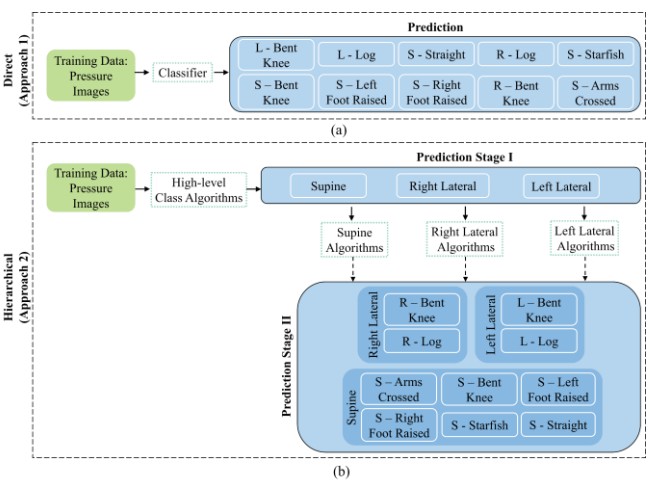

Fig. 3. Model schematics for (a) Approach 1 and (b) Approach 2.

## E. Hyper-Parameter Tuning

Five-fold cross validation was used to tune the hyper-parameters for the four 2D CNN models for both approaches. Four hyper-parameters were examined: batch size, number of iterations, number of epochs, and learning rate. We tried various batch sizes ranging from 20 to 70, three epochs:10, 20, and 30, three learning rates: 0.1, 0.001, and 0.0001, and iterations ranging from 20 to 120 to find the optimal value for each hyper-parameter that would produce the best validation accuracy. Fig. 4 shows box plots related to the validation accuracies for each 2D CNN algorithm. Fig. 4 (a) and (b) show the results of Approach 1 and Approach 2, respectively. We observe that the ShuffleNet algorithm consistently achieved high accuracies of over 95% for both approaches. The ResNet-18 algorithm showed more consistent behaviour compared to the AlexNet and GoogLeNet algorithms. Overall, Approach 1 provided more stable accuracy values compared to Approach 2. The hyperparameters used for the ViT-Large were taken from the ShuffleNet algorithm, due to high performance and stability.

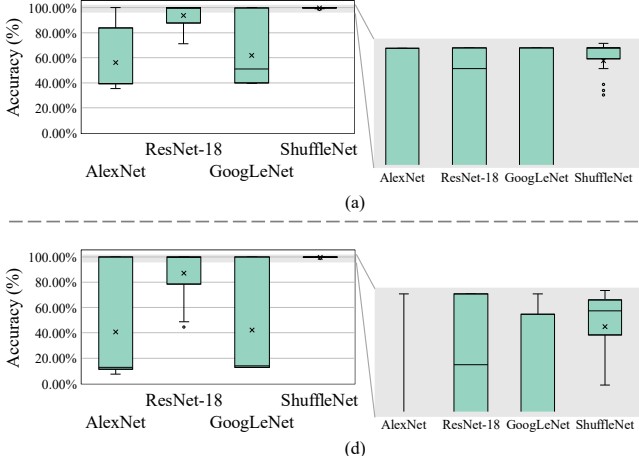

Fig. 4. Box plots representing the hyper-parameter tuning performance for all four 2D CNN algorithms using 5-fold for (a) Approach 1 and (b) Approach 2.

## III. EXPERIMENTAL RESULTS

The macro classification metrics, such as accuracy, sensitivity, specificity, and F1-Score, shown in the following equations, were calculated to demonstrate the performance of all five algorithms.

$$Accuracy = \frac{T_p + T_n}{T_n + T_p + F_n + F_p} \qquad Sensitivity = \frac{T_p}{T_p + F_n}$$

$$Specificity = \frac{T_n}{T_n + F_p} \qquad Precision = \frac{T_p}{T_p + F_p}$$

$$F1 - Score = 2 * \frac{Precision * Sensitivity}{Precision + Sensitivity}$$

Where $T_P$, $T_N$, $F_P$, and $F_N$ denote true positives, true negatives, false positives, and false negatives, respectively. Fig. 5 displays the classification metrics for all five algorithms in both approaches using holdout and LOSO cross validation. ShuffleNet was the algorithm that achieved the highest performance. For Approach 1, it obtained F1-scores of 99.66% and 92.92% ± 9.11% for holdout and LOSO cross validation, respectively. For Approach 2, it achieved F1-scores of 99.75% and 93.53% ± 7.37% for holdout and LOSO cross validation, respectively. This was to be expected as ShuffleNet is known to be an efficient and highly accurate algorithm to classify images [22]. The higher performance in Approach 2 could be due to the high performance achieved from the high-level class algorithm from [11], reducing inter-class misclassification.

The ResNet-18 algorithm achieved a better performance in Approach 1 compared to Approach 2, resulting in F1-Scores of 99.79% and 92.56% ± 9.38% for holdout and LOSO cross validation, respectively. Contrastingly, the GoogLeNet algorithm achieved a better performance in Approach 2, resulting in F1-Scores of 99.29% and 91.84% ± 7.05% for holdout and LOSO cross validation, respectively.

The AlexNet algorithm achieved a better performance when using the direct classification in Approach 1, resulting in F1-Scores of 99.72% and 90.99% ± 6.33% for holdout and LOSO cross validation, respectively. However, notably, the AlexNet algorithm also achieved the worst performance of all four 2D CNN models, with F1-Scores of 86.94% and 77.74%

± 26.54% for holdout and LOSO cross validation, respectively, when using Approach 2. These results may be attributed to the model's shallow network architecture, which might struggle to distinguish between each sub-posture.

Lastly, the ViT-Large algorithm achieved a better performance using Approach 1, resulting in F1-Scores of 100% and 90.01% ± 6.02% for holdout and LOSO cross validation, respectively. However, a reduction in performance occurred with Approach 2, resulting in F1-Scores of 99.18% and 71.59% ± 10.43% for holdout and LOSO cross validation, respectively. This discrepancies could be due to the varying dataset sizes in the two approaches, as this algorithm is known to perform worse on smaller datasets [24].

Overall, these results conclude that the performance of the direct classification and hierarchical architecture are dependent on the algorithm architecture. Additionally, it should be noted that the holdout technique consistently performed better than the LOSO technique, which is to be expected, as previous literature shows that the performance of an algorithm decreases when using LOSO [16] [17]. Lastly, the 2D CNN algorithms performed better than the ViT-Large algorithm, which could be due to the small and simple dataset as well as simpler and more efficient algorithms [24].

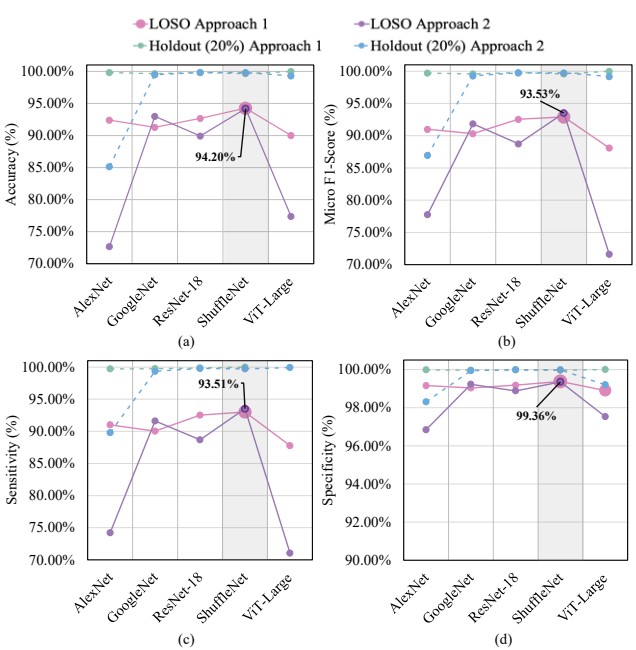

Fig. 5. Performance of all four 2D CNN and ViT-Large algorithms, with (a) Accuracy, (b) F1-Score, (c) Sensitivity, and (d) Specificity.

Fig. 6 depicts misclassification rates for all models using LOSO. The greatest misclassification occurred in the AlexNet and ViT-Large models when using Approach 2. For the AlexNet model, shown in Fig. 6 (b), most sub-postures are misclassified with the left bent knee (L-BentKnee) subclass shown in purple. Though this misclassification can be understood for the left log (L-Log), right bent knee (R-BendKnee), and right log (R-Long) subclasses, as these postures are all quite similar to each other, it is unclear why the algorithm confused the left bent knee sub-posture with any of the supine subclasses. For the ViT-Large model,

shown in Fig. 6 (j), most of the misclassification occurred within the supine high-level class. Typically, the misclassification occurred between similar postures, such as the arms crossed (S-ArmsCrossed), starfish (S-Starfish), and straight (S-Straight) postures, as well as the bent knee (S-BentKnee), right foot and left foot raised (S-RightFootRaised, S-LeftFootRaised) postures.

Contrastingly, the ShuffleNet models provided the lowest misclassification shown in Fig. 6 (g) and (h) for Approach 1 and Approach 2, respectively, indicating the models' stability. For almost all LOSO cross validation models, the supine arms crossed (S-ArmCrossed), and supine straight (S-Straight) subclasses primarily became misclassified with each other. This was anticipated, as these two sub-postures are very similar.

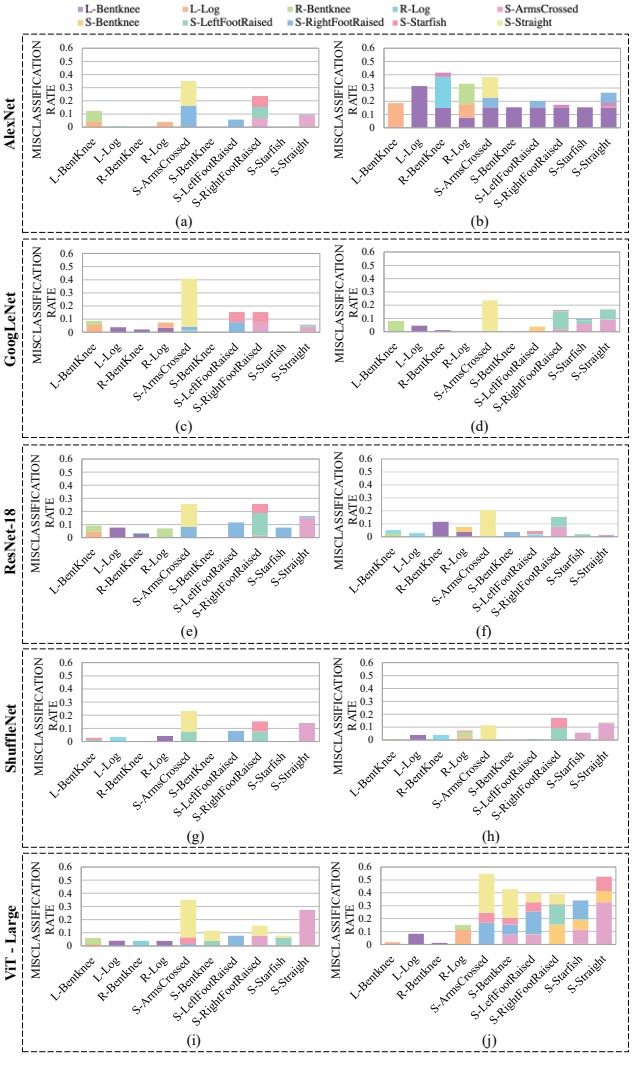

Fig. 6. Misclassification rate plots for: AlexNet (a) LOSO Approach 1 and (b) LOSO Approach 2; GoogLeNet (c) LOSO Approach 1 and (d) LOSO Approach 2; ResNet-18 (e) LOSO Approach 1 and (f) LOSO Approach 2; ShuffleNet (g) LOSO Approach 1 and (h) LOSO Approach 2; ViT-Large (i) LOSO Approach 1 and (j) LOSO Approach 2.

By examining the pressure distribution images, it was evident that in the supine arms-crossed sub-posture, some participants kept their triceps on the mattress when crossing their arms, while others removed all parts of their arms from the mattress. The images that often lead to this

misclassification were when participants chose to keep their triceps on the mattress as this creates a similar pressure distribution image to the supine straight subclass. The supine right foot raised subclass was often primarily misclassified as the supine left foot raised subclass. This could be due to the similarity in sub-postures with different legs raised off the mattress. Fig. 7 displays the performance of each model versus the heights and weights of the subjects. As shown in this figure, for both approaches, there is no clear correlation between model performance and the height/weight of the subject.

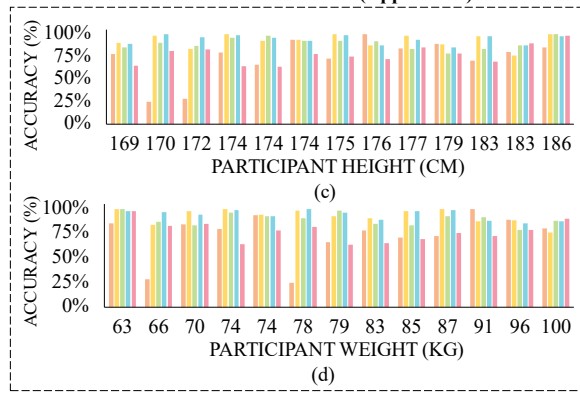

Fig. 7. Subject-specific performance plots of the 2D CNN and ViT-Large models corresponding to Approach 1 and (a) participant height, and (b) participant weight, Approach 2 and (c) participant height, and (d) participant weight.

To further evaluate the results, the gradient weights class activation map (Grad-CAM) technique was used. Grad-CAMs highlight the parts of an image that have the most impact on the classification through utilizing the gradients of the prediction scores. It is important to note that large gradient regions indicate places that most impact the final prediction scores. To better understand the highest misclassified subclasses in the ShuffleNet models, which include the supine arms crossed (S-ArmsCrossed), supine straight (S-Straight), and supine right foot raised (S-RightFoodRaised) subclasses, Grad-CAM images were developed for correctly and incorrectly classified images. Fig. 8 and Fig. 9 show the Grad-CAM images for Approach 1 and Approach 2, respectively. Fig. 8 (a) and (b) illustrate correctly, and incorrectly classified images of the S-ArmsCrossed subclass in Approach 1. In both images, it is evident that the model is primarily focusing on the upper limb

region to evaluate the presence of these limbs. Fig. 8 (c) and (d) display the correctly and incorrectly classified images of the S-Straight subclass. It appears that the correct focus should be on the whole body to classify this posture accurately. Focusing solely on the upper limb led to incorrect classification as S-ArmsCrossed. Fig. 9 (a)-(d) follow similar patterns. As mentioned previously, this model confusion could be explained through the presence of triceps placement on the mattress during the S-ArmCrossed subclass. Fig. 8 (e)-(g), and Fig. 9 (e)-(g) show that when the model accurately classifies the image, the focus of the model is around the torso and upper area of the lower limbs. However, when the model misclassifies these images with either supine left foot raised (S-LightFoodRaised) or supine starfish (S-Starfish), the model seems to focus more on the extremities rather than the torso placement.

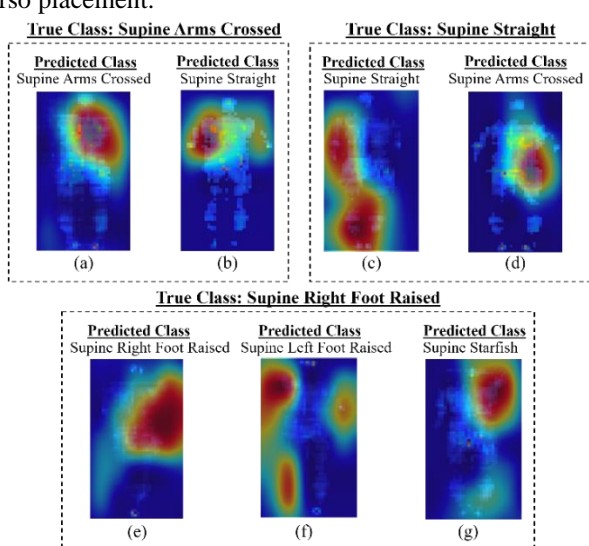

Fig. 8. Grad-CAM images for the ShuffleNet in approach 1. (a) correctly and (b) incorrectly classified images for the *supine arms crossed*, (c) correctly and (d) incorrectly classified images for the *supine straight* and (e) correctly and (f), (g) two incorrectly classified images for the *supine right foot raised* subclass.

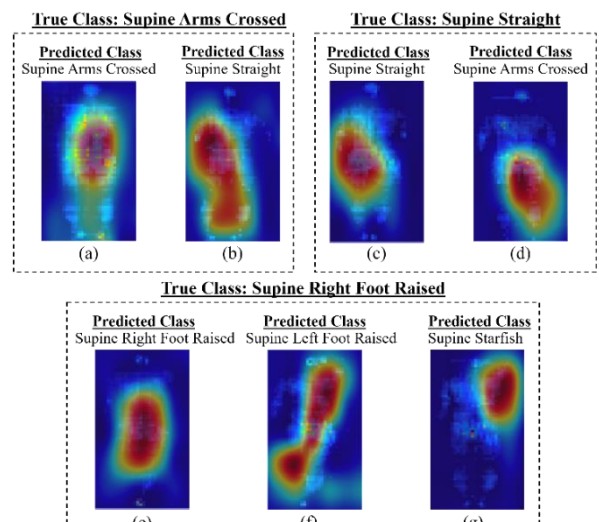

Fig. 9. Grad-CAM images for the ShuffleNet in approach 2. (a) correctly and (b) incorrectly classified images for the *supine arms crossed* subclass, (c) correctly and (d) incorrectly classified images for the *supine straight* subclass, (e) correctly and (f), (g) two incorrectly classified images for the *supine right foot raised* subclass.

Table II compares the previous literature that used smart mats to collect in-bed posture data to classify multiple sub-postures. The corresponding classification algorithms, cross validation methods and performance of each study is also provided. The accuracy of each study was reported in this table as this was a consitent metric reported by all studies. All the papers presented in Table II include sub-postures correlated to the supine, right lateral, and left lateral categories. However, the data within each paper was collected from varying subjects lying in various sub-postures on different smart mats, which could potentially impact the overall performance of the classification algorithms reported. Additionally, the classification algorithms evaluated within each study also vary, which could influence the overall reported performances. Therefore, it is essential to consider these factors when comparing the results of each study.

Overall, our study performed well compared to the other papers listed in Table II. Utilizing the holdout cross validation method allowed our model to achieve the highest performance in regards to classifying sub-postures. However, it is important to note that the performance of the ShuffleNet algorithm dropped by 5.63% when using the LOSO cross validation technique. [15] was the only other study that reported a subject specific cross validation method, which could explain the study's lower performance. Additionally, our proposed model factored in more specific sub-postures, totaling to 10 subclasses. Due to the high performance in our proposed models, this indicates that these models can generalize to more specific posture categories.

TABLE II.    A COMPARISON BETWEEN PREVIOUS LITERATIRE AND OUR PROPOSED MODELS

| Ref. | #Sub | #Postures | Algorithm | CV[1] | Accuracy (%) |
|---|---|---|---|---|---|
| [13] | 20 | 8 | KNN[2] | 10-fold | 97.10 |
| [14] | 5 | 6 | CNN[3] | Holdout | 91.24 |
| [15] | 14 | 6 | KNN + BEMD[4] | LOSO | 90.78 |
| **Ours** | **13** | **10** | **ShuffleNet** | **Holdout** **LOSO** | **99.83** **94.20** |

[1]CV: Cross Validation; [2]KNN: K-Nearest Neighbour; [3]CNN: Convolutional Neural Network; [4]BEMD: Body-Earth Mover's Distance.

Although all five algorithms achieved high performances regarding the classification of the 10 sub-postures, it is important to recognize some limitations. Firstly, this dataset was collected on healthy participants, with no history of pressure injuries, suggesting that this dataset may not accurately reflect the target population that is most likely to use a pressure injury monitoring system. Additionally, this dataset tasked participants to lie in supervised postures for a maximum of 2 minutes. This is not the most realistic use case as the goal of this in-bed posture classification system is to detect participant specific normalized sleeping postures. Furthermore, this portion of the dataset was only collected on a singular type of mattress, which does not accurately include all mattress types that are commonly used within the pressure injury risk population, such as therapeutic foam mattresses and hospital beds. Therefore, we need a dataset that includes patients at risk of developing pressure injuries, collected in an overnight setting, and encompassing all common

mattresses used for such patients. In the future, we plan to expand our dataset to address these limitations.

Furthermore, when using the hierarchical architecture (Approach 2) to classify the postures, only one type of neural network was used for both the high-level objective classification and the respective sub-posture classification. Combining the best-performing models at each phase within Approach 2 may lead to a higher performance that will be investigated in our future work. Lastly, Approach 1 provided a more stable model than Approach 2, requiring future modifications to Approach 2 to ensure reliability in real-time. Therefore, in the future it will be important to collect a more diverse dataset and explore various data augmentation techniques to address misclassification errors and reliability.

## IV. CONCLUSION

Four 2D CNN models and a Vision Transformer were used to classify 10 in-bed sub-postures, correlated to the supine, right lateral, and left lateral postures, using their pressure distribution images. Two approaches were used in this classification: a direct comparison (Approach 1) and a hierarchical evaluation (Approach 2). The ShuffleNet algorithm achieved the highest performance in Approach 2 with F1-Scores of 99.75% ± 1.43% and 93.53% ± 7.37% for holdout and LOSO cross validations, respectively. However, Approach 1 closely followed in performance, achieving F1-scores of 99.59% ± 0.34% and 92.92% ± 9.11% for holdout and LOSO cross validations. These models provide a good foundation for developing an automated in-bed posture classification algorithm. With a more diverse dataset, that includes PI risk patients, non-standardized sleeping postures, and various mattress types, an in-depth algorithm can be developed for practical use. Eventually, these models can be used as a preventative measure for PI development to notify caregivers when it is time to reposition the patient if the patient has not repositioned themselves naturally. Additionally, since full body pressure images are captured, high risk pressure regions can be highlighted to aid the caregivers in understanding the best posture to reposition the patient, thus offloading high risk regions, and decreasing PI development. Future considerations for practical use include the use of moisture-resistant sheets to protect the pressure mat and to maintain patient hygiene as well as investigation into a reminder mechanism to avoid notification overload on the caregivers and ensure comfort for the patients. It is important to co-design this system with users to ensure use and maintenance in clinical and non-clinical settings.

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
