# OpenReview forum: "Towards an AI-Based In-Bed Posture Detection System for Pressure Injury Prevention"
_IEEE.org/EMBS/BHI/2024/Conference — IEEE BHI'24_

### Official Review · Reviewer_ut6j · 2024-08-04
**Towards an AI-Based In-Bed Posture Detection System for Pressure Injury Prevention**

**Overall Rating:** 7
**Confidence:** 3

**Other Quality Metrics:**

- Clarity of writing: good
- Clinical Significance: great
- Methodological Novelty: good
- Experiments and Results: good

**Questions For The Authors:**

What motivates the division of the analyzes into the two approaches: “direct classification” and “hierarchical architecture”? Are both approaches equivalent? How do you explain the different results obtained?

**Strengths:**

The paper proposes AI-based techniques for evaluating images of in-bed body postures, which could be of practical use for the clinician to prevent pressure injuries in patients.
The paper is nicely written and understandable for readers of medical or computer science background.

The paper offers a strongly detailed introduction and review of the subject. A solid overview of the structure of the 4 studied convolution neural networks is providen.
The cross-validation algorithms holdout and LOSO were used and compared in the analysis and their functioning is well-explained.

Explainability of the methods is explored through class activation maps (with Grad-CAM), providing solid visualizations of the classifications on the posture pictures.

Analyzes of the four proposed CNN algorithms favored the use of the ShuffleNet tool which will be further improved on in the future.

**Summary Of The Paper:**

The paper compares the performance of four Convolution Neural Networks (CNN) systems on image classification of in-bed postures. Two cross-validation techniques, holdout and leave-one-subject-out (LOSO), and two approaches for classifying are considered and compared: “direct classification” and “hierarchical architecture”. The paper shows that the ShuffleNet algorithm achieves the highest performance out of the four CNN tools. The paper opens the road for further improving automation in the medical field of in-bed posture detection for preventing pressure injuries.

**Weaknesses:**

It was unclear to me how many postures (classes for classifying the pictures) are present in the dataset. Section A mentions 17 different postures, but 3 out of those are excluded, as explained in the text, so the remaining 14 are shown in Figure 1.

However, section D starts as: “Two different methods were used to classify the 10 postures”, and from this point on, the number of postures and classes gets mentioned as 10 in the rest of the text (+ abstract). As I understand, the 10 postures should result from the merge of the two classes “Log 30” and “Log 60” into “Log”, and “Bent knee” and “Fetus” into “Bent Knees”, eliminating 4 possible classes. However, this should be clarified in the text.

The division of classification problems into two approaches: “direct classification” (a one-step classification) and “hierarchical architecture” (a two-step classification) is also described in section D. I find the description of these two approaches to be lacking, considering these are central to the results, so I suggest the following:

- Move the part about merging classes from section D to section A so it is clear before section D that the data is simplified into 10 classes
- Expand the description in section D on the two approaches and explain why this division of the problem into two is necessary. Perhaps an illustrative schema could also be helpful.

There are typos in the mathematical definitions: Specificty → Specificity, Percision → Precision

---

### Official Review · Reviewer_6X2z · 2024-08-10
**This research demonstrates the promising potential of using pressure sensitive sheets and deep learning models to automatically monitor in-bed postures**

**Overall Rating:** 6
**Confidence:** 4

**Other Quality Metrics:**

(a) Clarity of writing: good
(b) Clinical Significance: Good
(c) Methodological Novelty: poor
(d) Experiments and Results: Great

**Questions For The Authors:**

The authors should try the attention-based classification model. AlexNet, GoogLeNet, ResNet-18, and ShuffleNet are more than 8 years old models.

**Strengths:**

The methodology is correct.
Achieve improved performance results.

**Summary Of The Paper:**

This research paper investigates the application of a pressure-sensitive sheet combined with convolutional neural networks (CNNs) to automatically identify in-bed body postures and prevent the development of pressure injuries. The study evaluates four CNN models—AlexNet, GoogLeNet, ResNet-18, and ShuffleNet.

**Weaknesses:**

The authors did not explore the recently developed model/algorithm.

---

### Official Review · Reviewer_ykTo · 2024-08-16
**Towards an AI-Based In-Bed Posture Detection System for Pressure Injury Prevention**

**Overall Rating:** 8
**Confidence:** 5

**Other Quality Metrics:**

Clarity of Writing: Good - The paper is well-organized and the concepts are generally well-explained, though some technical details could be elaborated further for clarity.
Clinical Significance: Great - The work addresses a significant clinical problem with the potential for real-world impact.
Methodological Novelty: Good - While the application of CNNs is not entirely new, the specific approach and the comparison of different architectures add value.
Experiments and Results: Great - The experimental design is thorough, with multiple validation techniques that strengthen the findings.

**Questions For The Authors:**

Dataset Diversity: Could you provide more details on the diversity of the subjects in the dataset (e.g., age, gender, body type)? How do you think these factors might impact the model's performance?
Hierarchical Model Stability: Given that direct classification provided more stable results, what improvements do you think are necessary for the hierarchical model to be more reliable?
Real-world Implementation: Have you considered any plans or collaborations for testing this system in actual healthcare environments? What challenges do you anticipate in such implementations?
Misclassification Strategies: What strategies or adjustments are you considering to reduce the misclassification rates, especially for similar sub-postures?

**Strengths:**

Clinical Relevance: The research addresses a critical issue in healthcare—preventing pressure injuries, which are both common and costly.
Innovative Application: The use of deep learning models, particularly CNNs, to analyze pressure distribution data is a novel approach that could advance the field of in-bed posture detection.
Comprehensive Evaluation: The paper thoroughly compares multiple CNN architectures and employs both holdout and leave-one-subject-out (LOSO) cross-validation techniques, providing a robust evaluation of the models.
Practical Implications: The development of a non-invasive, privacy-preserving system has practical implications for improving patient care in clinical settings.

**Summary Of The Paper:**

The paper proposes an AI-based in-bed posture detection system aimed at preventing pressure injuries (PIs) in patients with limited mobility. It utilizes a pressure-sensitive sheet placed under bedsheets to capture body postures, which are then classified using 2D Convolutional Neural Networks (CNNs). The authors evaluate four CNN architectures—AlexNet, GoogLeNet, ResNet-18, and ShuffleNet—across two classification approaches: direct classification and a hierarchical model. The results show that the ShuffleNet model, particularly when using a hierarchical approach, provides the highest accuracy and F1-scores, indicating promising potential for real-time posture detection that could assist caregivers in timely repositioning of patients to prevent PIs.

**Weaknesses:**

Generalizability: The dataset used was relatively small and specific, with only 13 subjects, which may limit the generalizability of the findings to broader populations.
Hierarchical Model Limitations: Although the hierarchical approach with ShuffleNet achieved high performance, the paper notes that direct classification provided more stable results, indicating potential limitations of the hierarchical method.
Misclassification Issues: The misclassification rates, particularly in the AlexNet model, suggest that some CNN architectures struggle to differentiate between similar sub-postures, which could impact the system's reliability in real-world applications.
Real-world Testing: The system has not yet been tested in real-world clinical settings, which is crucial for validating its effectiveness and usability.

---

### Decision · Program_Chairs · 2024-09-23

Accept